# Contact network analysis of Covid-19 in tourist areas——Based on 333 confirmed cases in China

**Zhangbo Yang[1,2], Jingen Song[1], Shanxing Gao[3], Hui Wang[3]\*, Yingfei Du[4], Qiuyue Lin[1]**

**1** School of Humanities and Social Science, Xi'an Jiaotong University, Xi'an, China, **2** Institute for Empirical Social Science Research, Xi'an Jiaotong University, Xi'an, China, **3** School of Management, Xi'an Jiaotong University, Xi'an, China, **4** Zhou Enlai School of Government, Nankai University, Xi'an, China

\* whh609@163.com

**Data Availability Statement:** All data and code are on a ICPSR open data repository (DOI: 10.3886/E145901V2). All data have been anonymized. None of the experiments was preregistered.

## Abstract

The spread of infectious diseases is highly related to the structure of human networks. Analyzing the contact network of patients can help clarify the path of virus transmission. Based on confirmed cases of COVID-19 in two major tourist provinces in southern China (Hainan and Yunnan), this study analyzed the epidemiological characteristics and dynamic contact network structure of patients in these two places. Results show that: (1) There are more female patients than males in these two districts and most are imported cases, with an average age of 45 years. Medical measures were given in less than 3 days after symptoms appeared. (2) The whole contact network of the two areas is disconnected. There are a small number of transmission chains in the network. The average values of degree centrality, betweenness centrality, and PageRank index are small. Few patients have a relatively high contact number. There is no superspreader in the network.

## 1 Introduction

Coronavirus Disease 2019 (Covid-19) has broken out in many countries around the world and drawn widespread global attention since 2019 [1, 2]. On January 30, 2020, the World Health Organization declared the epidemic a Public Health Emergency of International Concern. This virus spreads via respiratory droplets, close proximity interactions (CPIs), and airborne transmission [3, 4], with an incubation period of 1–14 days. According to the joint investigation report of COVID-19 released by China and World Health Organization in February 2020, there is little or no pre-existing immunity in most populations [5]. Its main clinical manifestations include fever, dry cough, and fatigue. The outbreak of the epidemic has caused a major impact on society and economy, especially in areas where the tourism industry is more developed.

Existing research on COVID-19 is mostly based on epidemiology, virology, and medicine, involving patients' case analysis, infection model construction, gene sequencing, and clinical diagnosis, etc. [6–12]. However, there is a lack of analysis on virus transmission from the perspective of a complex network. In particular, existing models lack the description of particular diffusion paths from actor to actor, so the interventions based on these models lack pertinence [13]. The spread of the virus is closely related to human social network [14]. It is the

**Funding:** This research was supported by National Natural Science Foundation of China (Grant No. 71902155) and National Philosophy and Social Sciences Foundation of China (Grant No. 20VYJ052). The funders had no role in study design, data collection and analysis, decision to publish, or preparation of the manuscript.

connection, gathering, and movement of people, that makes the virus spread and form a transmission network [14–19]. Existing studies have analyzed the relationship between population movement among cities and infection ratio during the epidemic based on mobile phone data on the macro-level [20]. But there is still less analysis from the micro-individual level based on the contact network of COVID-19 patients. The topology of the contact network plays a critical role in virus spread [21]. Human social networks are heterogeneous, and the number of people an individual contact with every day is different. Therefore, the probability that each individual may spread the virus is not the same. Intervention policies should also be targeted and heterogeneous [22]. Prior research showed that the immunization strategy based on the actual contact network is better than the uniform random immunization strategy [19].

This study collects detailed information about patients diagnosed with COVID-19 that are reported by the Health Commission of Hainan and Yunnan Provinces of China in Janauary and February, 2020. These two regions are major tourism provinces in China, so the epidemic has a greater impact on the local economy and society. We extract patients' demographics and characteristics by text mining method, and construct the contact network based on patient movement tracks and contact records.

## 2 Data and method

All samples in this study are officially announced cases by the Health Commission of Hainan and Yunnan Provinces of China. By February 16, 2020, there are a total of 333 cases. The study received approval from the Ethics Committee of Xi'an Jiaotong University Health Science Center (No. 2020–1217). All the data are anonymous. The specific steps of text mining and coding are as follows.

First, based on the detailed information of confirmed cases, we summarized these patients' demographic and epidemiological characteristics, including gender, age, residence, place of onset, place of infection, time of arrival, symptom outbreak time, quarantine time, and diagnosis time, etc.

Second, the Health Commissions of Hainan and Yunnan listed whether the patient is a family member, friend, or close contact of the previously confirmed patients. We construct the patient contact network in chronological order, calculate relevant network indicators, and draw a dynamic visual network of virus transmission between patients to show how the epidemic develops day by day. We use three network indicators to measure the position of patients in the network.

The first statistic we consider is the degree centrality. It measures the number of direct contacts of one patient [16]. The specific formula is as following, where $a_{ij}$ is whether case i contacts case j.

$$C_{AD}(i) = \sum_{j=1}^{n} a_{ij}$$

Our next statistic is betweenness centrality. It measures whether one patient is in the bridge position of a transmission path in the network [16]. The higher the betweenness centrality of a patient, the higher the risk of him or her spreading the virus. The specific formula is as followin. The betweenness centrality of case i is $g_{jk}(i)$ divided by $g_{jk}$, which means the ratio of the geodesic passing through case i and connecting case j and k to the total number of geodesic between case j and k.

$$C_{AB(i)} = \frac{g_{jk}(i)}{g_{jk}}$$

Then we consider PageRank index. It measures the centrality of a patient in the whole network [23, 24]. PageRank measures how likely one is to arrive at a given node by moving randomly around a network, and is calculated iteratively. In the calculating process, we not only consider the number of direct contacts of patients but also the number of indirect contacts. The PageRank value of all nodes in the network converge to limiting values as the number of iterations goes to infinity. More details of the calculation can be found in [24]. Cases with a higher PageRank value are at the center of the whole contact network. The specific formula is as follows. $PR_j$ is the PageRank value that the case j directly contacted with the case i, $n_j$ is case j's ego network size, which means the number of cases connected with case j, and $B_i$ is the set containing all nodes linking to the focal node j.

$$PR_i = \sum_{j \in B_i} \frac{PR_j}{n_j}$$

Finally, we visualize the network by decomposing it into different communities based on components separated layout. A component means that, within in it, all members can be connected by a well-defined path. Besides, no connection exists between members from two different components. Through visualization, we can intuitively observe changes in the epidemic contact network over time.

## 3 Results and discussion

### 3.1 Epidemiological characteristics of patients with COVID-19

Based on the movement trajectories of confirmed cases reported by the Health Commission of Hainan and Yunnan Provinces, basic information of the patients' gender, place of infection, and route of infection were counted (see Table 1). Specifically, there are slightly more female cases, accounting for more than half. Over 80% cases were infected out of the province, which means that most cases are imported. Also, about 48% cases' relatives were infected.

Average age of all cases in these two areas is 45 years old, with the youngest 3 months old, and the oldest 79 years old. The frequency distribution of cases' age is left-skewed (see Fig 1). The most highly infected groups are the 25–40 years old and the 50–70 years old, among which the 65-year-old has the largest number, with 11 people infected.

As showed in Figs 2 and 3, cases' average age in the two provinces varies with time. In the data of Hainan Province, except for January 27, January 28, and February 9, the average age is stable at 40–60 years old. In the data of Yunnan Province, except February 4, February 8, and February 12, the average age is stable at 30–50 years. Compared with Hainan Province (48.44

**Table 1. Epidemiological information of cases with COVID-19 in Hainan and Yunnan Province of China (N = 333)*.**

| Gender | Frequency | Percentage (%) | Infected place | Frequency | Percentage (%) |
|---|---|---|---|---|---|
| Female | 142 | 50.35 | Inside Province | 55 | 18.71 |
| Male | 140 | 49.65 | Outside Province | 239 | 81.29 |
| Total | 282 | 100 | Total | 294 | 100 |
| **Whether any relatives are infected?** | Frequency | Percentage (%) | **Age** | Mean | S.D. |
| Yes | 106 | 47.75 | | 45.33 | 18.06 |
| No | 116 | 52.25 | | | |
| Total | 222 | 100 | | | |

* In Yunnan Province, 51 cases did not report gender. In the two provinces, 39 cases did not report places of infection, 111 cases did not report contacts, and 52 cases did not report age.

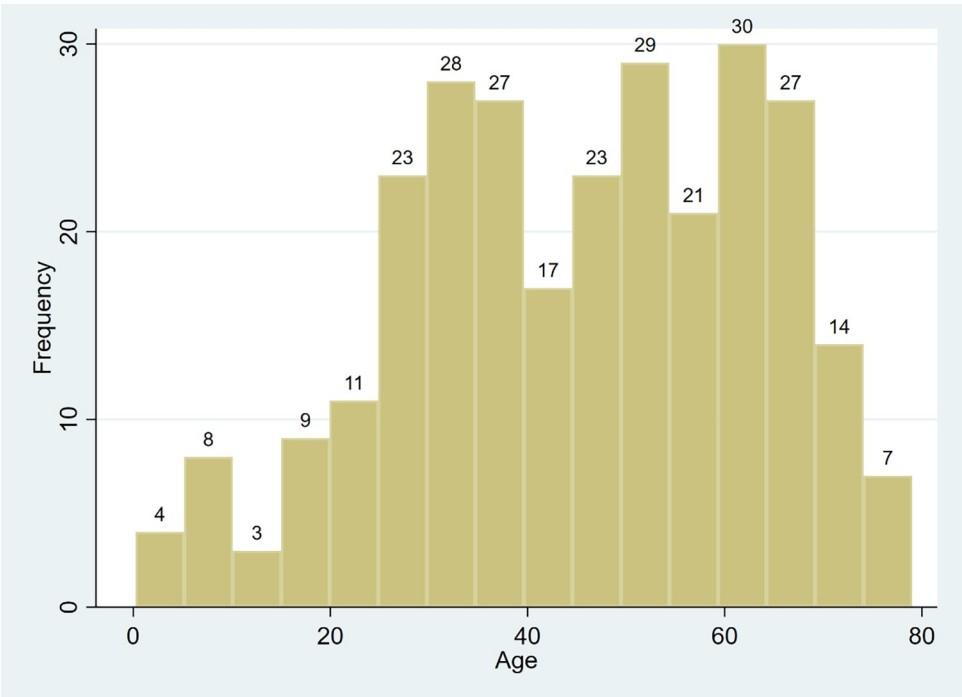

**Fig 1. Age distribution of Covid-19 cases in Hainan Province and Yunnan Province, China.** (N = 281).

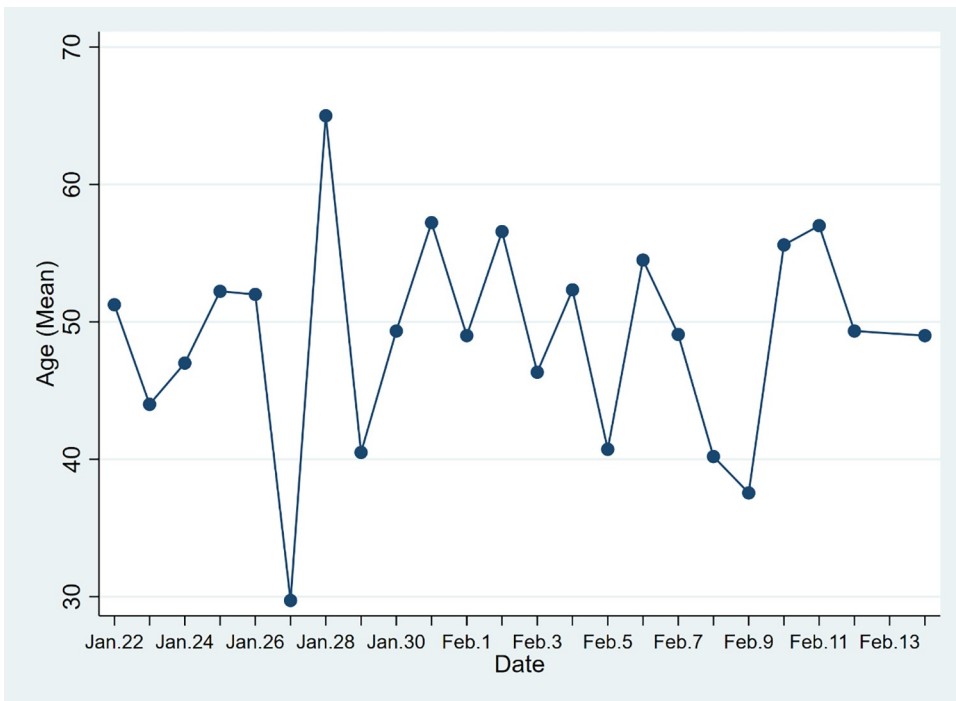

**Fig 2. Mean Age of Covid-19 cases in Hainan Province, China.**

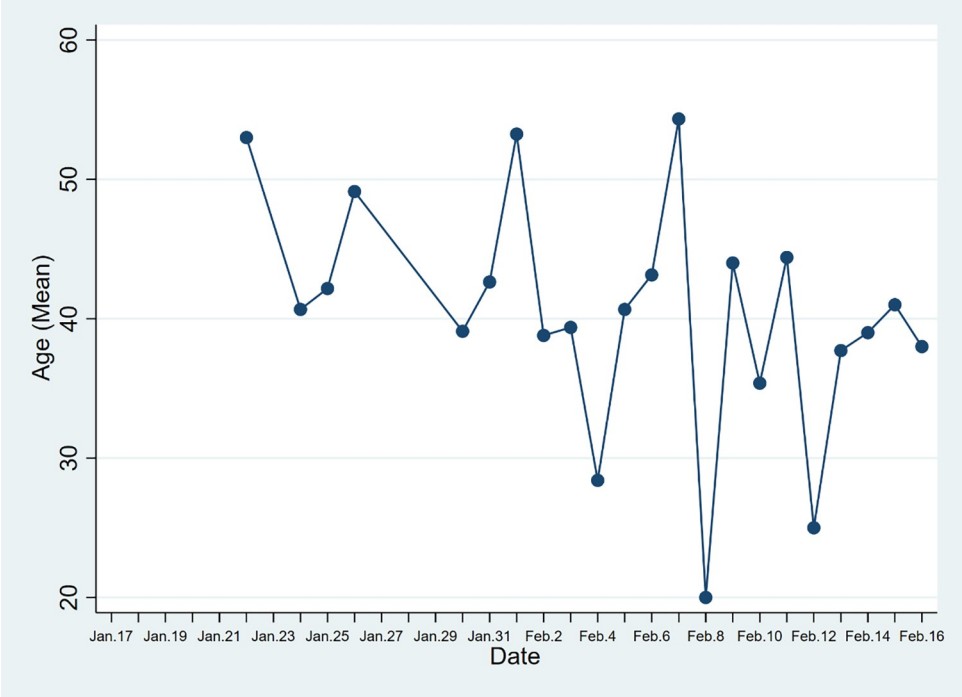

**Fig 3. Mean Age of Covid-19 cases in Yunnan Province, China (cases before Jan.22 did not report patient's age).**

years), the average age of cases in Yunnan Province (41.09 years) is lower. One possible reason for the age gap might be that the tourism industry is the pillar industry in Hainan Province, mild and warm all-year-round climate attracts more elderly tourists coming to spend the winter. Tourism in Yunnan Province is typically less concentrated towards older age groups. In Yunnan Province, 21.01% of the cases are aged 60 years and above, which exceeds the proportion of the local population aged 60 years and above in the total population (14.91%, China the seventh National Census). In Hainan Province, 32.72% of the cases are aged 60 years and above, which is greater than the proportion of this age group in the total population (14.65%, China the seventh National Census). Compared with young people, older people may have a weaker immune system, which will make them more susceptible to the virus.

For imported cases, we also summarized the number of days between arrival in Hainan or Yunnan and emergence of symptoms (such as cough, fever, etc.), the number of days between arrival and relevant medical measures(such as seeing a doctor, quarantine, etc.), and the number of days between emergence of symptoms and taking medical measures. As showed in Table 2, statistics show that in Hainan, the average time between arrival and emergence of

**Table 2. Statistics related to the onset time of Covid-19 cases in Hainan and Yunnan Province, China.**

| Variables | Number of cases | Mean | S.D. | Minimum value | Maximum value |
|---|---|---|---|---|---|
| Symptom onset date minus arriving in **Hainan** date (days) | 23 | 9.96 | 15.73 | -1 | 66 |
| Diagnosis/quarantined date minus arriving in **Hainan** date (days) | 118 | 9.92 | 17.18 | -11 | 118 |
| Diagnosis/quarantined date minus Symptom onset date (days) in **Hainan** | 31 | 2.58 | 3.40 | -4 | 12 |
| Symptom onset date minus arriving in **Yunnan** date (days) | 6 | 4.50 | 13.58 | -19 | 22 |
| Diagnosis/quarantined date minus arriving in **Yunnan** date (days) | 40 | 8.05 | 5.17 | 1 | 23 |
| Diagnosis/quarantined date minus Symptom onset date (days) in **Yunnan** | 31 | 2.55 | 5.49 | -2 | 23 |

symptoms is 9.96 days, with the earliest case emerge symptoms 1 day before arrival and the latest 66 days after arrival. The average time between arrival and seeing a doctor or being quarantined is 10 days, with the earliest case taken medical treatment 11 days before arrival. The average time between the emergence of symptoms and seeing a doctor or taking medical treatment is 3 days. The earliest case took medical measures 4 days before onset of symptoms, and the latest one took medical measures 12 days after onset of symptoms. While in Yunnan Province, the data is a bit different. The average time between arrival and emergence of symptoms is 4.5 days in Yunnan, with the earliest case emerge symptoms 19 days before arrival and the latest one 22 days after arrival. The average time between arrival and seeing a doctor or being quarantined is 8 days. The earliest case went to see a doctor once arriving, whereas the latest one did not take any medical treatment until 23 days after arriving in Yunnan. The average time of taking relevant medical treatment after symptoms emerge is about 2 days.

It is worth noting that there are lots of imported cases in Hainan and Yunnan. Many of them are tourists, and they have a large range of activities. For example, cases No. 73 and No. 87 in Hainan Province crossed districts and counties about 8 times. Such movement tracks increased the risk of spreading the virus and difficulty for epidemic prevention.

Figs 4 and 5 respectively show the average symptoms onset time and average diagnosis time per day in each province. In both regions, the average diagnosis time and onset time have an increasing trend. This means that with the development of the epidemic, it takes more time to discover potential cases.

## 3.2 Contact network of patients with COVID-19

We visualized the dynamic contact network of patients with COVID-19 in Hainan and Yunnan provinces by February 16, 2020. As showed in Figs 6 and 7, each figure contains three

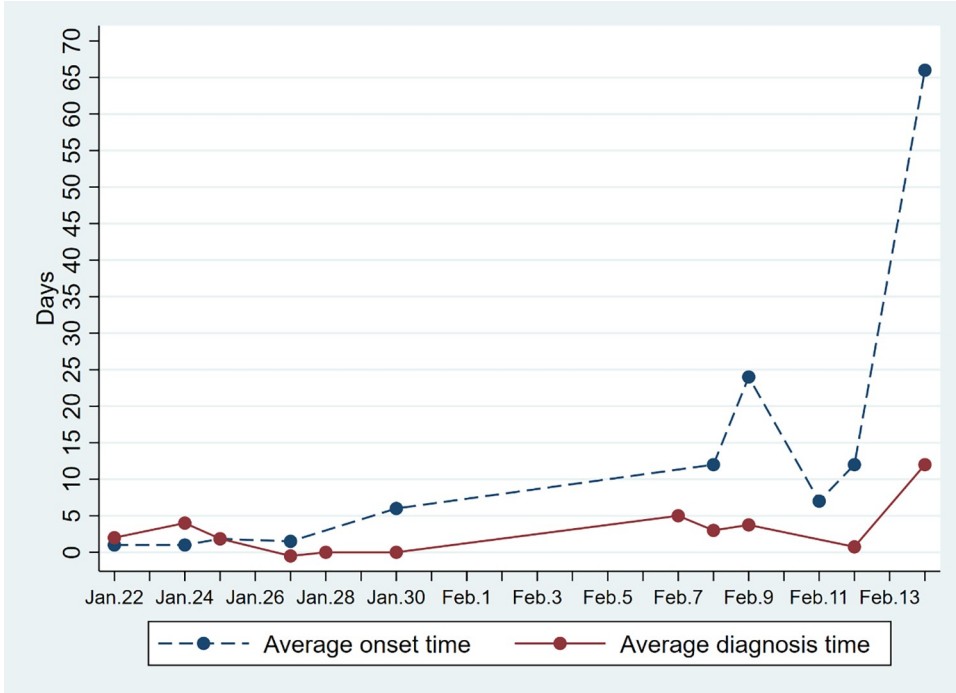

**Fig 4. Changes in the average onset and diagnosis time of Covid-19 cases in Hainan Province.** Average onset time refers to the time when symptoms appear in imported cases after they arrive in Hainan. Average diagnosis time refers to the time of taking medical measures after symptoms appear.

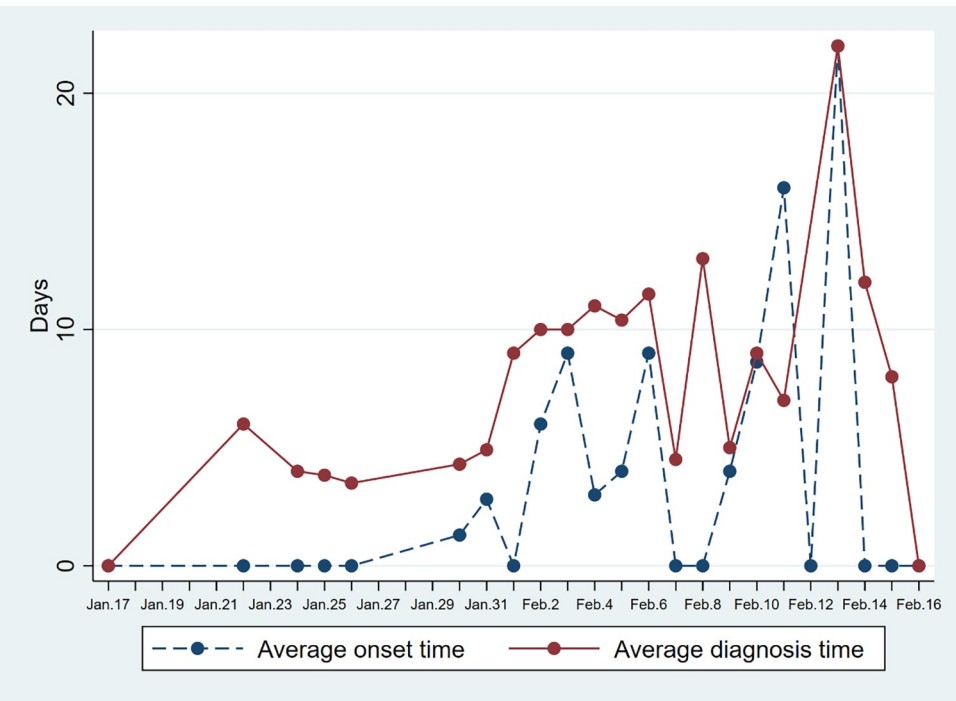

**Fig 5. Changes in the average onset and diagnosis time of Covid-19 cases in Yunnan Province.** Average onset time refers to the time when symptoms appear in imported cases after they arrive in Yunnan. Average diagnosis time refers to the time of taking medical measures after symptoms appear.

parts: the network at the early stage of the epidemic, the network at the middle stage of the epidemic, and the network at the late stage of the epidemic. Nodes represent cases, numbers in nodes are diagnosis date. The larger the number is, the later that case was diagnosed. Edges in the network represent contact relation (family members, friends, close contact, or strangers) between cases. Size of node represents the degree centrality of each case. The larger the node is, the more contact relations that case has with others. There are many unconnected nodes in each network, most of which are imported cases. According to existing information, it is hard to trace the source of infection out of province.

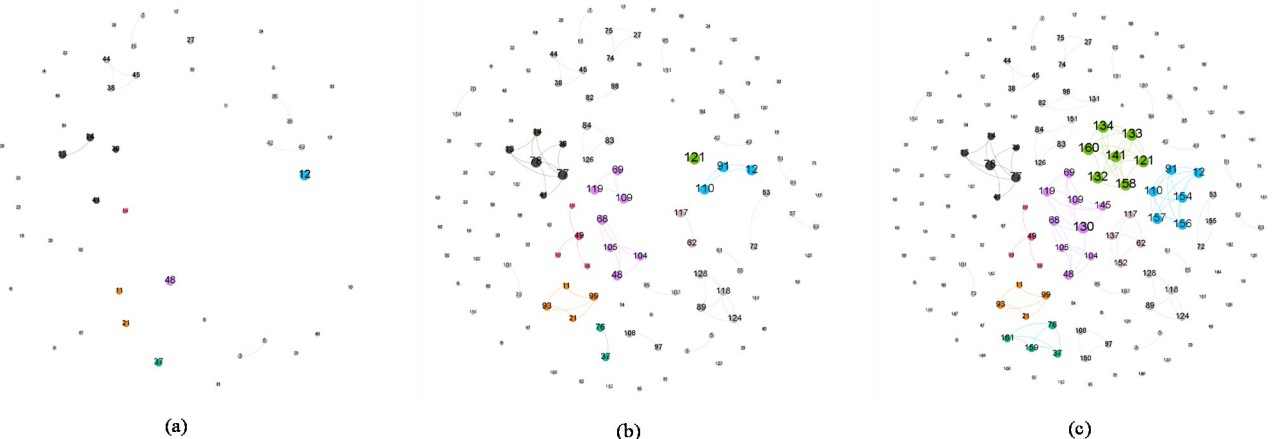

**Fig 6.** Dynamic contact network of cases with Covid-19 in Hainan (a) January 30, 2020; (b) February 8, 2020; (c) February 16, 2020.

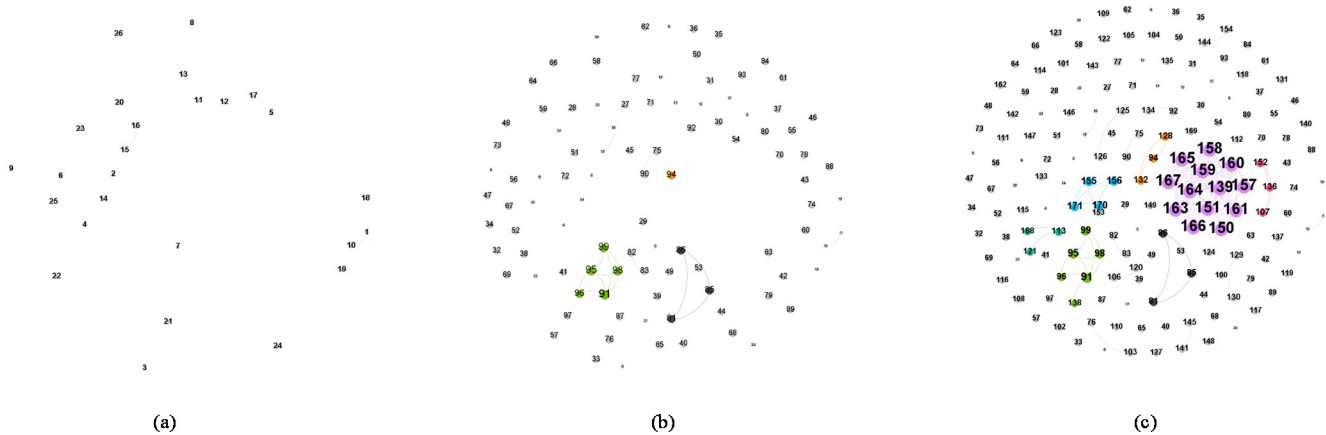

**Fig 7.** Dynamic contact network of cases with Covid-19 in Yunnan (a) January 30; (b) February 8; (c) February 16.

Fig 6 shows the contact networks in Hainan Province. The early network by January 30 is still relatively sparse, most cases no relation with others. Cases No.38, No. 44, and No. 45 are from one family and formed the largest cluster (component). They are imported cases from Wuhan, Hubei Province. In the mid-term network by February 8, several major infection clusters are formed, most of which are cluster infections caused by imported cases (tourists from Hubei). The size of these clusters is normally 3–6, and would still be expanding in the late stage. In the late stage network by February 16, the newly emerged cluster is composed of 7 cases including case No. 121, which is a local family infection cluster. The largest cluster consists of 9 cases including case No. 130. There are many clusters of full connection in the network, which are composed of family members or close friends. According to the contact data, we found that there are more parallel infection networks caused by strong connections (cases in the network are diagnosed almost at the same time) than serial infection networks caused by strangers or weak connections (cases in the network have a clear order of diagnosis).

It can be seen from Fig 7 that the early network in Yunnan Province was also relatively sparse. There were only two clusters in the first stage network, both did not expand further. It indicates that these cases were effectively controlled at the early stage and did not cause further spread. Cases in the two clusters were infected in Wuhan and traveled to Yunnan. In the mid-term, cluster infections appear, several major clusters emerged in the network. Cases in these clusters are mostly imported cases that traveled from Wuhan to Yunnan for visiting family members. Some of their close contacts in Yunnan were also infected. However, these clusters did not form larger clusters in the late stage, indicating that most infected people were given medical treatment or quarantined in time. The later network was divided into multiple clusters, most of which were fully connected. Among them, the largest cluster is composed of thirteen cases including case No.139. In this cluster, case No.139 participated in a village-wide gathering activity and contacted with people returned from Hubei, leading to this cluster of infection. In the network, case No.139 is a typical hub of virus spreading and should be the primary target of epidemic prevention. In general, compared with the contact network in Hainan, the contact network in Yunnan has fewer clusters. However, there is a very large cluster, which is mainly caused by case No.139. This shows that the epidemic prevention and control in the tourist area should focus on preventing imported super-infected persons.

Table 3 shows the centrality indicators of the contact network in Hainan Province. Degree centrality can reflect the basic reproduction number of the virus in a certain extent. The average degree centrality of cases with COVID-19 in Hainan Province is 1.51. The minimum

**Table 3. Statistics of the contact network of Covid-19 cases in Hainan Province.**

| Variables | Cases | Mean | S.D. | Minimum Value | Maximun value |
|---|---|---|---|---|---|
| Degree Centrality | 162 | 1.51 | 1.79 | 0 | 6 |
| Betweenness Centrality | 162 | 0.0000 | 0.0001 | 0.0000 | 0.0012 |
| PageRank Index | 162 | 0.0062 | 0.0043 | 0.0015 | 0.0189 |

centrality is 0, and the maximum is 6. The maximum centrality is case No.121, who connected with other six cases. These cases formed a fully connected cluster. Betweenness centrality indicates whether the patient is in a bridge position of the infection chain. Its average value is 0.0000, indicating that there are fewer infection chains formed in the contact network. PageRank index indicates the centrality of each case's position in the whole contact network rather than ego network. The average value is only 0.0062, while the maximum value is 0.0189, indicating that the degree of connection between cases is unevenly distributed and the deviation is large.

Table 4 shows the network indicators of Yunnan Province. The average degree centrality is 1.345. The smallest degree is 0 and the largest is 12. The average value is smaller than that of Hainan Province(1.51), but the standard deviation is much larger. That is because there is one case (Case No.139 in Fig 7) that owned 12 contacts and they formed the largest cluster in the network. The betweenness centrality of the contact network in Yunnan Province is same of Hainan Province, with an average of 0.0000. The main reason is that there are many clusters in the network, and there is no path between these clusters. Therefore, the overall connectivity of the network is at a low level. The average PageRank index of Yunnan Province is 0.0058, which is lower than that of Hainan Province (0.0062), but the standard deviation is larger than that of Hainan Province. The maximum value (0.0221) is also larger than that of Hainan Province (0.0015). This indicates that a small number of cases own more contacts than others, leading to the more uneven distribution than Hainan Province.

Figs 8 and 9 show the trend of network indicators over time, including the number of nodes, the number of edges, the average degree, the density, and the size of the largest component. In both regions, the number of nodes, the number of edges, and the average degree show an obvious increasing trend. This means that with the development of the epidemic, the number of cases and the connection between them are increasing. But the density and the size of the largest component did not change a lot. It indicates that the density of the contact network in these two areas is both at a low level, and the size of the components is mostly on a small scale.

## 4 Conclusion

By analyzing two major tourist regions of Hainan and Yunnan in China, the study found that: (1) The susceptible population is mostly middle-aged men. There were more imported cases in the early stage of the epidemic and more local cluster infections in the late stage. Cases in the two regions received medical treatment in less than three days on average after they developed

**Table 4. Statistics of the contact network of Covid-19 cases in Yunnan Province.**

| Variables | Cases | Mean | S.D. | Minimum Value | Maximum value |
|---|---|---|---|---|---|
| **Degree Centrality** | 171 | 1.345 | 3.187 | 0 | 12 |
| **Betweenness Centrality** | 171 | 0.0000 | 0.0000 | 0.000 | 0.0030 |
| **PageRank index** | 171 | 0.0058 | 0.0055 | 0.0020 | 0.0221 |

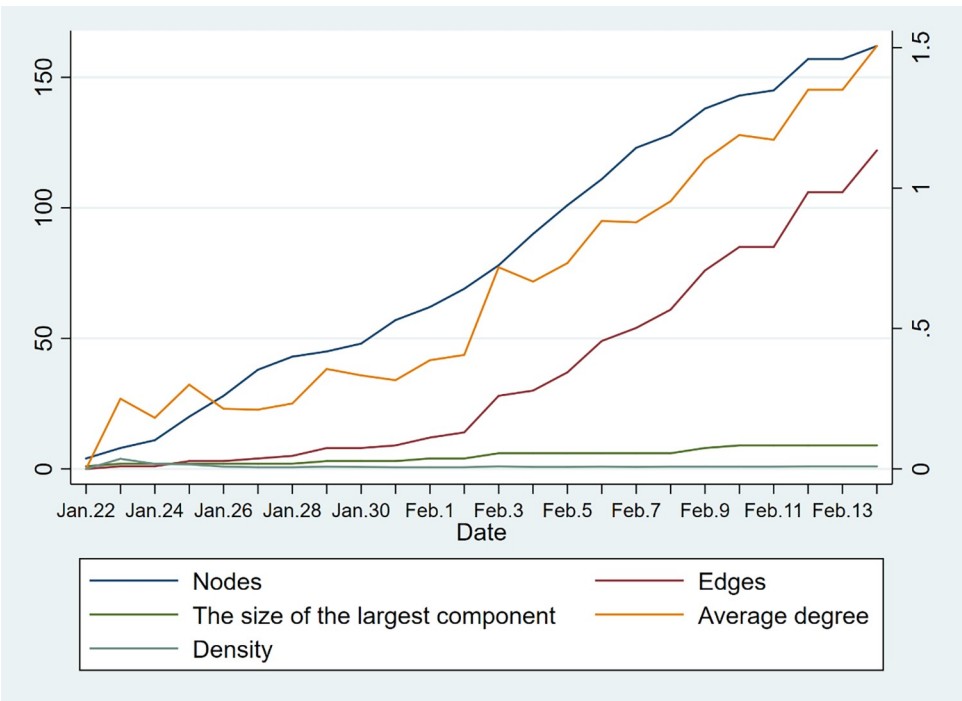

**Fig 8. Trend of network indicators of Hainan Province (Nodes, edges, and the size of the largest component refer to the y-axis coordinates on the left.** Average degree and density refer to the y-axis coordinates on the right).

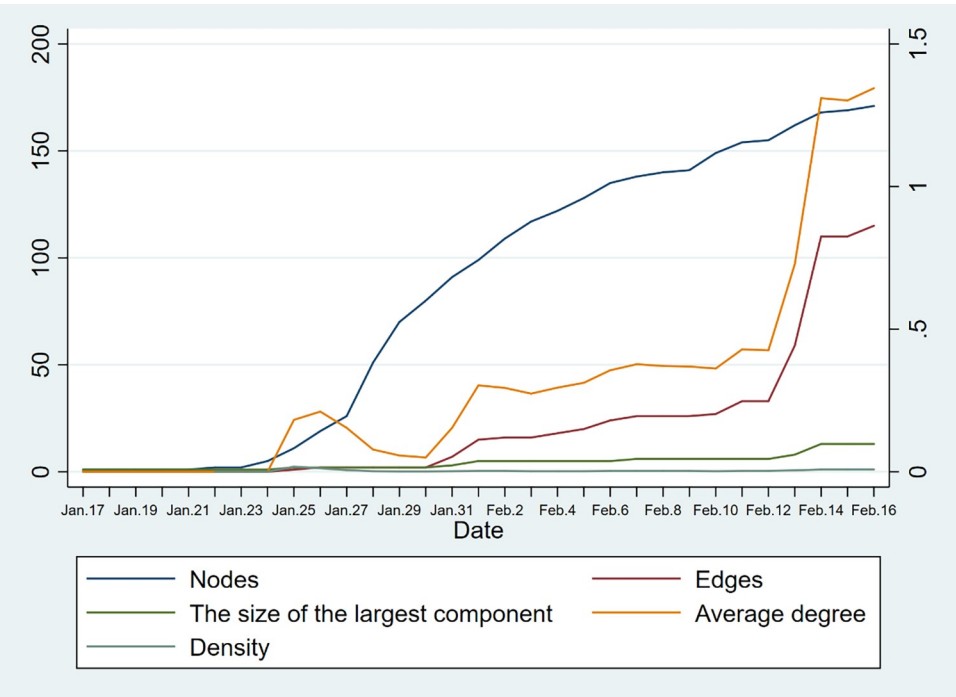

**Fig 9. Trend of network indicators of Yunnan Province (Nodes, edges, and the size of the largest component refer to the y-axis coordinates on the left.** Average degree and density refer to the y-axis coordinates on the right).

symptoms. In the early stage of the epidemic in the two provinces, cases are mostly young and middle-aged people who have a wide range of activities and strong transmission ability. At the late stage, cases are mostly elderly people. Our results have some reference value for future infectious disease control policies. However, specific epidemic prevention policies need to consider multiple data sources and be justified by rigorous interventions. (2) The contact network in two areas is consisted of multiple clusters. The whole network is disconnected, and the degree distribution is skewed. The highest contact number is 6 in Hainan and 12 in Yunnan, which shows that the network connectivity in these two areas is relatively low.

We did not include any intervention in this study and epidemic control policies in Hainan and Yunnan are essentially the same, so we cannot compare the effects of different political interventions. However, our study still illustrates that if we face with a similar epidemic situation in the future, abundant information can be mined from case report text for epidemic control. When this information is quickly collected and analyzed by epidemic prevention department, they can, after anonymization, provide the public with quick and clear understanding of virus transmission information such as age, gender, and onset time distribution of each case. The epidemiology department can also map the transmission network of virus based on merging multiple cases, allowing the public to understand the spread of the virus. This data analyzation and visualization are totally based on case reports, which can be easily implemented without much cost.

In general, most of the research on the COVID-19 epidemic has focused on the epidemiological characteristics of confirmed patients. This study helps to better understand the social network of COVID-19 cases by visualizing dynamic contact networks and analyzing network centrality. Due to the difficulty of data collection and lack of patient movement information provided by the Health Commission, we only analyzed confirmed cases. We don't know much about those imported cases before their arrival. Information such as movement trajectory and infection route is still unclear. Therefore, it is hard to present a cross-regional case contact network. Furthermore, Covid-19 can also be spread via airborne transmission [3, 4], especially in more confined spaces [25]. Under such conditions, transmission may occur between two people that are in the same space while did not meet face to face, so network analysis cannot fully capture the virus transmission pathway. Because of limited data, we underestimated the corresponding network metrics and connectivity. Therefore, a more complete analysis of virus transmission paths requires the integration of network analysis, geography information system and mobile trajectory analysis. The above needs to be further studied based on more detailed case information.

## Acknowledgments

We thank for Joe Hilton's insightful comments. We also thank for Editor Federico Botta's help.

## Author Contributions

**Conceptualization:** Zhangbo Yang.

**Formal analysis:** Zhangbo Yang, Jingen Song, Yingfei Du.

**Supervision:** Shanxing Gao.

**Visualization:** Zhangbo Yang, Jingen Song, Yingfei Du.

**Writing – original draft:** Zhangbo Yang, Jingen Song, Yingfei Du.

**Writing – review & editing:** Zhangbo Yang, Shanxing Gao, Hui Wang, Qiuyue Lin.

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
