## [Decision Letter · Decision Letter 0]

18 Jun 2021

PONE-D-21-13537

Contact Network Analysis of Covid-19 in Tourist Areas——Based on 333 Confirmed Cases in China

PLOS ONE

Dear Dr. Wang,

Thank you for submitting your manuscript to PLOS ONE. After careful consideration, we feel that it has merit but does not fully meet PLOS ONE’s publication criteria as it currently stands. Therefore, we invite you to submit a revised version of the manuscript that addresses the points raised during the review process.

Ensure that all comments raised by the reviewer have been addressed. Particular care should be given to the comments about the policy implications and make sure that the results are not overstated in the context of the manuscript.

We look forward to receiving your revised manuscript.

Kind regards,

Federico Botta

Academic Editor

PLOS ONE

Journal Requirements:

4. Please ensure that you refer to Figure 9 in your text as, if accepted, production will need this reference to link the reader to the figure.

Reviewers' comments:

Reviewer's Responses to Questions

**Comments to the Author**

1. Is the manuscript technically sound, and do the data support the conclusions?

Reviewer #1: Partly

2. Has the statistical analysis been performed appropriately and rigorously? 

Reviewer #1: I Don't Know

3. Have the authors made all data underlying the findings in their manuscript fully available?

Reviewer #1: No

4. Is the manuscript presented in an intelligible fashion and written in standard English?

Reviewer #1: No

5. Review Comments to the Author

Reviewer #1: The authors have carried out an analysis of epidemiological data gathered in two Chinese provinces during the early stages of the COVID-19 pandemic, including an analysis of the age and gender profile of cases as well as of an inferred underlying transmission network. While the analysis itself has been conducted to a good standard, I have a few comments about the presentation of the work, and I am particularly concerned that the authors have overstated the policy implications of their analysis.

Main comments:

As stated in the short summary of my review, the authors have repeatedly made statements on policy which are not justified by the scientific content of the paper. The authors have studied the epidemiological characteristics of a relatively small number of cases and analysed the underlying contact network, and have not carried out any mechanistic modelling of control measures or interventions. I am not suggesting that the authors should be carrying out mechanistic modelling, but they need to remove, or at least qualify, some of their statements on policy. In the abstract, the authors say “The study suggests that the tourism industry should adopt a strategy of opening up different scenic spots during the epidemic and strengthen the detecting and tracking of tourists”. In general, when making policy recommendations it is much better to state results in the form “If policy X is implemented, then our study suggests outcome Y will occur”, rather than just “Policy X should be implemented.” The “If X then Y” phrasing is more information-rich and states a scientific finding, whereas the “X should be implemented” phrasing contains less information and states a political, or even moral, imperative. The “If X then Y” phrasing is thus better suited to the abstract of a scientific paper. However, since the authors have not performed any modelling of interventions or drawn explicit connections between the differences in policy in the two provinces and the differences in epidemic spread, I do not feel their work justifies any definite “If X then Y” statements; while their analysis may point to places where a certain policy might be useful, it certainly is not capable of rigorously predicting the impact of a given policy. In particular, I see no way the authors can predict the impact of opening up certain scenic spots based on the work presented here. I would also like to point out that this is a very vague policy recommendation, and based on what is stated in the abstract and in the conclusions it could refer to any control measures other than the closure of all scenic spots.

In their conclusion section on page 21, the authors say “It shows that prevention and control measures such as quarantine are effective”. First of all, it is not entirely clear what the “It” here is – I assume they are referring to their complete analysis, but it would be good to make this clear. More importantly, the only detailed reference to quarantine measures comes in their analysis of the Yunan transmission network on page 16, where they say “these clusters did not form larger clusters in the late stage, indicating that the infected people were taken medical treatment or quarantined in time”. There appears to be some circular reasoning at work here: the authors infer that small cluster size has been caused by quarantining without explaining why they have made this inference, and then later argue that their analysis supports the use of quarantine. However, based on the evidence provided one could (very facetiously) propose any mechanism one wanted to for the small cluster size, and then later argue that the analysis supports policies involving this mechanism. The authors need to either remove the comment about quarantine on page 21, or provide more concrete reasoning for why they think their analysis supports the use of quarantine.

On similar grounds, I recommend that the authors remove the entire paragraph on page 22 beginning “Based on the research results, we should strictly control the number of tourists…” and ending “…registration of information”. No indication is given of what the impact of the measures discussed in this paragraph is likely to be, or how this potential impact relates to the results of the network analysis. The authors could replace this paragraph with a more qualified discussion of the implications of their work for control policy.

Overall I recommend that the authors completely rewrite all of the sections of their paper which talk about policy interventions. While I do not necessarily disagree with any of these statements, I do not believe that the analysis conducted in the paper is at all sufficient to support the very confident recommendations which the authors make. A justified policy recommendation would need to state the likely impact of the policy, and the work presented here is not capable of predicting such impacts. A more tentative discussion of policy, outlining the ways the authors’ analysis could support a policy response, would be very welcome, but the sweeping recommendations made in the current manuscript would require either a mechanistic modelling study or a direct comparison of data from regions implementing different policies to be justified.

In their introduction, the authors state “This virus spreads via respiratory droplets and close proximity interactions (CPIs), with an incubation period of 1-14 days.“ While there has been some controversy on this point I believe it is now well-established that COVID-19 can be spread through airborne transmission (I have provided a reference for this at the end of this review, but I recommend the authors seek out some more references to get a wider view of the evidence on this topic). This transmission pathway is obviously less determined by human social networks than the CPI and droplet components. While a network analysis is still extremely worthwhile, it would be good to point out the limitations implied by airborne transmission; for instance, infection can potentially spread across large rooms or between neighbouring rooms connected by a ventilation system, meaning the index and secondary case may not necessarily have directly interacted and so will not necessarily belong to a shared social contact network. While the authors are already careful not to imply that all transmissions are “on-network” transmissions, statements like “The analysis of the virus contact network is the key to understand and clarify the spread of disease” may be overstating the importance of the transmission network in light of “non-network” airborne transmissions.

On page 6, the authors introduce some network statistics, stating “These are the three specific network indicators that we used to depict the postion of patients in the network”. I am not sure that “depict” is the right word here. These statistics really act to quantify the patients’ positions, and I would like to see this sentence adjusted accordingly. As an aside, I note here that “position” is misspelt.

The definition of betweenness centrality on page 6 is incomplete. In their verbal explanation the authors say “The betweenness centrality of the patient i is the ratio of the geodesic passing through patient i and connecting patients j and k to the total number of geodesic between patients j and k.” This clearly isn’t a correct description; based on the description given on Wikipedia, I believe this should be the sum of all such ratios over all distinct pairs of j and k. In their algebraic formula they do not define the term b_jk(i). If this term is supposed to be the ratio, the authors should state this. If it is just meant to be the number of geodesics passing through I, then they also need to add an appropriate denominator to their formula. Similarly, the authors do not provide any indication of how to calculate Pagerank index. The authors need to either state how it can be calculated, or provide an appropriate reference. The Brin and Page 1998 paper does not actually include an explicit formula for Pagerank and so is not sufficient here. The authors also mention “patient j’s ego network size” in this section, but do not explain what an ego network is. For publication, a definition of the ego network needs to be provided.

The description of the network component on page 7 is a bit confusing: “Network component means that within it, each patient has a network path reach with others, it may be a direct tie or an indirect tie”. I think what the authors mean is that all members of a network component can be connected by a well-defined path. It would be good to reword this sentence to make its meaning clearer.

In Figure 1 on page, no indication is given of what the solid line corresponds to. I assume this is a distribution which has been fitted to the data. The authors do not actually appear to use this fitted distribution anywhere in the paper, and no a priori reasoning is given for why the unimodal distribution they have attempted to fit should give a good approximation of the age distribution of cases. In practice the age distribution of cases will depend on a range of factors including the age distribution of the underlying population, the contact patterns within that population, and age-stratified heterogeneities in behaviour and physiology. This complex combination of factors means that a simple parametric distribution is unlikely to give a good explanation of the data, and so I do not think there is anything to be gained from fitting to such a distribution. I therefore recommend that the authors either remove this line from the plot, or provide a description of their fit and a compelling explanation as to why they have carried it out. The choice of bins and axis ticks in Figure 1 also needs to be changed. In the body text they refer to five-year intervals, their bars appear to be defined according to four-year intervals, and the axis ticks are at five year intervals starting at 3. The figure needs to be redrawn with five-year bins starting at zero, and matching axis ticks. In the body text describing Figure 1 the authors say “The frequency distribution of patients’ age is the left-skewed distribution”. I believe they mean that the frequency distribution is left-skewed, since “the left-skewed distribution” is not any specific probability distribution.

Typos and minor comments:

Abstract - “The spread of virus is highly related to the stucture of human network.” should be “The spread of virus is highly related to the structure of human networks.”

“There are more male patients than the females in these two districts, most are imported cases, with an average age of 45 years” should be “There are more male patients than the females in these two districts and most are imported cases, with an average age of 45 years”

“There is no superspreader in the network” – since outbreaks can have more than one superspreader, this should really say “There are no superspreaders in the network”.

Page 2: “Coronavirus Disease 2019 (Covid-19) has outbroken in many countries around the world and drew widespread global attention” should be “Coronavirus Disease 2019 (Covid-19) has broken out in many countries around the world and drawn widespread global attention”

Page 3: “Population is generally susceptible.” This statement is a bit vague. Something like “There is little or no pre-existing immunity in most populations”, with an appropriate citation, would be more specific.

Page 5: “Pior research showed that the immunization strategy based on the actual contact network is better than the random immunization strategy ( Eubank et al., 2004).” While I am not familiar with the Eubank et al. paper, a network-based immunisation strategy can still be random in the sense that each individual has a certain probability of being vaccinated, and I expect the authors mean uniform random immunisation (i.e. each individual has the same probability of being immunised) rather than just random immunisation. The authors have also misspelt “prior” at the start of this sentence.

Pages 6-7: The list of network statistics (degree centrality, betweenness centrality, Pagerank index) is currently presented as three paragraphs, with the first sentence of each being the name of the statistic. Depending on what the journal’s formatting requirements allow, I would rather see this presented in either a more explicit list form (i.e. with bullet points or at least bold item headings) or as “normal” text (i.e. “The first statistic we consider is…. Our next statistic is…. Finally, we consider….”).

Page 7: “Over 80% of patients were infected out of the province, which means there are more imported cases.” While the meaning here is clear, this is slightly poor wording, and should be amended to something like “Over 80% of patients were infected out of the province, which means that most cases are imported.”

Table starting on page 7 spills over to the next page and should be reformatted.

Page 9: “Hainan Province, where is a major pure tourism area in China” should be “Hainan Province, which is a major pure tourism area in China”

Table 2 on page 11: the column labelled “Case number” should be labelled “Number of cases”.

Tables 3 and 4 have final column labelled “Maximus value”, which should be “Maximum value”.

Page 22: “In general, most of the researches on the COVID-19 epidemic…” should be “In general, most of the research on the COVID-19 epidemic…”

Data availability – while the authors have provided a link to a repository containing their data and code, I was unable to access it. Under the “Files” tab on the repo website I got a message saying “Loading files” which did not resolve itself after >10 minutes of waiting. Because I can not see the code and the authors have also not provided a complete methodology for their calculation of betweenness centrality and Pagerank index, I can not confirm that their analysis has been carried out correctly.

Reference for airborne transmission:

Lidia Morawska, Julian W. Tang, William Bahnfleth, Philomena M. Bluyssen, Atze Boerstra, Giorgio Buonanno, Junji Cao, Stephanie Dancer, Andres Floto, Francesco Franchimon, Charles Haworth, Jaap Hogeling, Christina Isaxon, Jose L. Jimenez, Jarek Kurnitski, Yuguo Li, Marcel Loomans, Guy Marks, Linsey C. Marr, Livio Mazzarella, Arsen Krikor Melikov, Shelly Miller, Donald K. Milton, William Nazaroff, Peter V. Nielsen, Catherine Noakes, Jordan Peccia, Xavier Querol, Chandra Sekhar, Olli Seppänen, Shin-ichi Tanabe, Raymond Tellier, Kwok Wai Tham, Pawel Wargocki, Aneta Wierzbicka, Maosheng Yao,

How can airborne transmission of COVID-19 indoors be minimised?,

Environment International,

Volume 142,

2020,

105832,

ISSN 0160-4120,

https://doi.org/10.1016/j.envint.2020.105832.

6. PLOS authors have the option to publish the peer review history of their article (what does this mean?). If published, this will include your full peer review and any attached files.

Reviewer #1: **Yes: **Joe Hilton

---

## [Author Response · Author response to Decision Letter 0]

11 Aug 2021

Authors’ Response

Dear Editor and Review,

Thank you very much for the helpful comments that you and the reviewer provided on our manuscript. We are very pleased with the detailed modification suggestions and grateful to have the opportunity to refine our paper on these valuable suggestions. As you will see, we have addressed all the issues raised, which are highlighted in yellow in the manuscript and detailed below point by point. We look forward to your rechecking the paper and making other valuable suggestions for us. 

Thank you once more for your constructive feedback and tireless patience. We hope that the refined manuscript meets your expectations.

Yours sincerely,

Authors of PONE-D-21-13537

Main comments: Please ensure that you refer to Figure 9 in your text as, if accepted, production will need this reference to link the reader to the figure.

Our response: Thank you very much for your attentiveness. We have checked the text and corrected this typo to Figure 8.

Main comments: As stated in the short summary of my review, the authors have repeatedly made statements on policy which are not justified by the scientific content of the paper. The authors have studied the epidemiological characteristics of a relatively small number of cases and analysed the underlying contact network, and have not carried out any mechanistic modelling of control measures or interventions. I am not suggesting that the authors should be carrying out mechanistic modelling, but they need to remove, or at least qualify, some of their statements on policy.

Our response: Thanks for your suggestions. We have removed all statements on policy because we can’t do any intervention.

Main comments: In their conclusion section on page 21, the authors say “It shows that prevention and control measure such as quarantine are effective”. First of all, it is not entirely clear what the “It” here is – I assume they are referring to their complete analysis, but it would be good to make this clear. More importantly, the only detailed reference to quarantine measures comes in their analysis of the Yunan transmission network on page 16, where they say “these clusters did not form larger clusters in the late stage, indicating that the infected people were taken medical treatment or quarantined in time”. There appears to be some circular reasoning at work here: the authors infer that small cluster size has been caused by quarantining without explaining why they have made this inference, and then later argue that their analysis supports the use of quarantine. However, based on the evidence provided one could (very facetiously) propose any mechanism one wanted to for the small cluster size, and then later argue that the analysis supports policies involving this mechanism.

Our response: According to your suggestions, we removed related discussion on policy measures. We note that our findings can be used as a reference in the policy process, but it is clear that specific policies require a combination of multiple data sources. 

Main comments: Overall I recommend that the authors completely rewrite all of the sections of their paper which talk about policy interventions. While I do not necessarily disagree with any of these statements, I do not believe that the analysis conducted in the paper is at all sufficient to support the very confident recommendations which the authors make. A justified policy recommendation would need to state the likely impact of the policy, and the work presented here is not capable of predicting such impacts. A more tentative discussion of policy, outlining the ways the authors’ analysis could support a policy response, would be very welcome, but the sweeping recommendations made in the current manuscript would require either a mechanistic modelling study or a direct comparison of data from region simplementing different policies to be justified.

Our response: We have completely rewritten the section on policy. We did not revisit the effectiveness of the policy because this was not possible in the absence of interventions and regional comparisons. We mainly discussed the fact that epidemic prevention departments can quickly analyze case texts, integrate case data, map networks, and make them available to the public after anonymization in the case of similar disease outbreaks. This can give the public a clear picture of development of the outbreak.

Main comments: In their introduction, the authors state “This virus spreads via respiratory droplets and close proximity interactions (CPIs), with an incubation period of 1-14 days.“ While there has been some controversy on this point I believe it is now well-established that COVID-19 can be spread through airborne transmission (I have provided a reference for this at the end of this review, but I recommend the authors seek out some more references to get a wider view of the evidence on this topic). This transmission pathway is obviously less determined by human social networks than the CPI and droplet components. While a network analysis is still extremely worthwhile, it would be good to point out the limitations implied by airborne transmission; for instance, infection can potentially spread across large rooms or between neighbouring rooms connected by a ventilation system, meaning the index and secondary case may not necessarily have directly interacted and so will not necessarily belong to as hared social contact network.

Our response: In addition to the literature given by the reviewer, we also refer to some other literature related to airborne transmission. In the introduction section, we added airborne as a main transmission pathway. In the conclusion section, we added a discussion about the mode of transmission. Airborne transmission makes network analysis cannot depict the full transmission as you pointed out since virus can transmit between two strangers in different space, and the virus transmission should be discussed in the context of geographical location and space. We also put it into the limitation of this study.

Main comments: On page 6, the authors introduce some network statistics, stating “These are the three specific network indicators that we used to depict the postion of patients in the network”. I am not sure that “depict” is the right word here. These statistics really act to quantify the patients’ positions, and I would like to see this sentence adjusted accordingly. As an aside, I note here that “position” is misspelt.

Our response: Thank you very much for your suggestions for corrections. We have adjusted “depict” to “measure” and modified the misspelt of “position”. New sentence is as: “These are the three specific network indicators that we use to measure the position of patients in the network.”

Main comments: The definition of betweenness centrality on page 6 is incomplete. In their verbal explanation the authors say “The betweenness centrality of the patient i is the ratio of the geodesic passing through patient i and connecting patients j and k to the total number of geodesic between patients j and k.” This clearly isn’t acorrect description; based on the description given on Wikipedia, I believe this should be the sum of all such ratios over all distinct pairs of j and k. In their algebraic formula they do not define the term b_jk(i). If this term is supposed to be the ratio, the authors should state this. If it is just meant to be the number of geodesics passing through I, then they also need to add an appropriate denominator to their formula.

Our response: Thank you very much for your suggestions. We have revised the calculation formula of the betweenness centrality by adding an appropriate ratio to our formula to make it clearer and more explicit.

Main comments: Similarly, the authors do not provide any indication of how to calculate Pagerank index. The authors need to either state how it can be calculated, or provide an appropriate reference. The Brin and Page 1998 paper does not actually include an explicit formula for Pagerank and so is not sufficient here. The authors also mention “patient j’s ego network size” in this section, but do not explain what an ego network is. For publication, a definition of the ego network needs to be provided.

Our response: Thank you very much for your corrections. We have stated how Pagerank index can be calculated and provide an appropriate reference. And we also have provided a definition of the ego network. In our study, the patient j’s ego network size means the number of patients connecting with patient j.

Main comments: The description of the network component on page 7 is a bit confusing: “Network component means that within it, each patient has a network path reach with others, it may be a direct tie or an indirect tie”. I think what the authors mean is that all members of a network component can be connected by a well-defined path. It would be good to reword this sentence to make its meaning clearer.

Our response: Thank you very much for your suggestions. We have corrected the description of the network component on page 7 to make its meaning clearer. Network component means that within it, each patient has a network path reach with others, it may be a direct tie or an indirect tie, all members of a component can be connected by a well-defined path.

Main comments: In Figure 1 on page, no indication is given of what the solid line corresponds to. I assume this is a distribution which has been fitted to the data. The authors do not actually appear to use this fitted distribution anywhere in the paper, and no a priori reasoning is given for why the unimodal distribution they have attempted to fit should give a good approximation of the age distribution of cases. In practice the age distribution of cases will depend on a range of factors including the age distribution of the underlying population, the contact patterns within that population, and age-stratified heterogeneities in behaviour and physiology. This complex combination of factors means that a simple parametric distribution is unlikely to give a good explanation of the data, and so I do not think there is anything to be gained from fitting to such a distribution. I therefore recommend that the authors either remove this line from the plot, or provide a description of their fit and a compelling explanation as to why they have carried it out. The choice of bins and axis ticks in Figure 1 also needs to be changed. In the body text they refer to five-year intervals, their bars appear to be defined according to four-year intervals, and the axis ticks are at five year intervals starting at 3. The figure needs to be redrawn with five-year bins starting at zero, and matching axis ticks. In the body text describing Figure 1 the authors say “The frequency distribution of patients’ age is the left-skewed distribution”. I believe they mean that the frequency distribution is left-skewed, since “the left-skewed distribution” is not any specific probability distribution.

Our response: Thank you very much for your correction. We have redrawn Figure 1 by removing the solid line and changing axis ticks into five-year intervals starting at zero. We also revised the “left-skewed distribution” sentence: “The frequency distribution of patients’ age is left-skewed (see Figure 1).”

Typos and minor comments: Abstract - “The spread of virus is highly related to the stucture of human network.” should be “The spread of virus is highly related to the structure of human networks.”

Our response: Thank you very much for your corrections. We have changed this sentence in the abstract as: “The spread of virus is highly related to the structure of human networks”.

Typos and minor comments: “There are more male patients than the females in these two districts, most are imported cases, with an average age of 45 years” should be “There are more male patients than the females in these two districts and most are imported cases, with an average age of 45 years”.

Our response: Thank you very much for your corrections. We have modified the sentence into “There are more male patients than the females in these two districts and most are imported cases, with an average age of 45 years”.

Typos and minor comments: “There is no super spreader in the network” – since outbreaks can have more than one super spreader, this should really say “There are no super spreaders in the network”.

Our response: Thank you very much for your corrections. We have revised the sentence into “There are no super spreaders in the network”.

Typos and minor comments: Page 2: “Coronavirus Disease 2019 (Covid-19) has outbroken in many countries around the world and drew widespread global attention” should be “Coronavirus Disease 2019 (Covid-19) has broken out in many countries around the world and drawn widespread global attention” 

Our response: Thank you very much for your corrections. We have amended the sentence into “Coronavirus Disease 2019 (Covid-19) has broken out in many countries around the world and drawn widespread global attention”.

Typos and minor comments: Page 3: “Population is generally susceptible.” This statement is a bit vague. Something like “There is little or no pre-existing immunity in most populations”, with an appropriate citation, would be more specific.

Our response: Thank you very much for your corrections. We have revised the sentence into “According to the joint investigation report of COVID-19 released by China and World Health Organization in February 2020, there is little or no pre-existing immunity in most populations”.

Typos and minor comments: Page 5: “Pior research showed that the immunization strategy based on the actual contact network is better than the random immunization strategy ( Eubank et al., 2004).” While I am not familiar with the Eubank et al. paper, a network-based immunisation strategy can still be random in the sense that each individual has a certain probability of being vaccinated, and I expect the authors mean uniform random immunisation (i.e. each individual has the same probability of being immunised) rather than just random immunisation. The authors have also misspelt “prior” at the start of this sentence.

Our response: Thank you very much for your corrections. We do focus on uniform random immunization rather than just random immunization, so we clarified the expression and modified misspelt word as following: “Prior research showed that the immunization strategy based on the actual contact network is better than the uniform random immunization strategy.”

Typos and minor comments: Pages 6-7: The list of network statistics (degree centrality, betweenness centrality, Pagerank index) is currently presented as three paragraphs, with the first sentence of each being the name of the statistic. Depending on what the journal’s formatting requirements allow, I would rather see this presented in either a more explicit list form (i.e. with bullet points or at least bold item headings) or as “normal” text (i.e. “The first statistic we consider is…. Our next statistic is…. Finally, we consider….”).

Our response: Thank you very much for your corrections. We have revised the first sentence at the beginning of the paragraph on Pages 6-7 into unified and standardized format: “The first statistic we consider is degree centrality.” “Our next statistic is betweenness centrality.” “Then we consider Pagerank index.”

Typos and minor comments: Page 7: “Over 80% of patients were infected out of the province, which means there are more imported cases.” While the meaning here is clear, this is slightly poor wording, and should be amended to something like “Over 80% of patients were infected out of the province, which means that most cases are imported.”

Table starting on page 7 spills over to the next page and should be reformatted.

Our response: Thank you very much for your corrections. We have amended the sentence on Page 7 into “Over 80% of patients were infected out of the province, which means that most cases are imported”. We also reformatted Table 1 for the spilling over problem.

Typos and minor comments: Page 9: “Hainan Province, where is a major pure tourism area in China” should be “Hainan Province, which is a major pure tourism area in China”

Table 2 on page 11: the column labelled “Case number” should be labelled “Number of cases”.

Our response: Thank you very much for your corrections. We have revised the sentence into “Hainan Province, which is a major pure tourism area in China” on Page 9. And we also have changed the column of Table 2 on page 11 into “Number of cases”.

Typos and minor comments: Tables 3 and 4 have final column labelled “Maximus value”, which should be “Maximum value”.

Our response: Thank you very much for your corrections. We have modified the word “Maximus” into “Maximum” in Tables 3 and 4.

Typos and minor comments: Page 22: “In general, most of the researches on the COVID-19 epidemic…” should be “In general, most of the research on the COVID-19 epidemic…”.

Our response: Thank you very much for your corrections. We have modified the sentence into “In general, most of the research on the COVID-19 epidemic…”.

Data availability – while the authors have provided a link to a repository containing their data and code, I was unable to access it. Under the “Files” tab on the repo website I got a message saying “Loading files” which did not resolve itself after >10 minutes of waiting. Because I can not see the code and the authors have also not provided a complete methodology for their calculation of betweenness centrality and Pagerank index, I can not confirm that their analysis has been carried out correctly.

Our response: We are sorry about the improper link. We re-uploaded our research data on a new data repository ICPSR. Please access the deposit workspace at: https://www.openicpsr.org/openicpsr/project/145901/version/V2/view. The data and code are in a Stata format and the network file is in a Gephi file, you may need relevant software to open them. Apologize again for the inconvenience we have brought to your work.

Our response: Moreover, we asked a copy editor to help improve our language. Thank you again for your very detailed and extremely constructive comments, they help improve this paper a lot. We would not have got a high-quality paper without these sincere suggestions. We look forward to hear more about your feedback.

---

## [Decision Letter · Decision Letter 1]

13 Sep 2021

PONE-D-21-13537R1Contact Network Analysis of Covid-19 in Tourist Areas——Based on 333 Confirmed Cases in ChinaPLOS ONE

Dear Dr. Wang,

Thank you for submitting your manuscript to PLOS ONE. After careful consideration, we feel that it has merit but does not fully meet PLOS ONE’s publication criteria as it currently stands. Therefore, we invite you to submit a revised version of the manuscript that addresses the points raised during the review process.

In particular, you should carefully address the comments about the calculation of the PageRank index, as well as fixing the typographical and grammatical mistakes highlighted.

We look forward to receiving your revised manuscript.

Kind regards,

Federico Botta

Academic Editor

PLOS ONE

Journal Requirements:

Additional Editor Comments (if provided):

Reviewers' comments:

Reviewer's Responses to Questions

**Comments to the Author**

1. If the authors have adequately addressed your comments raised in a previous round of review and you feel that this manuscript is now acceptable for publication, you may indicate that here to bypass the “Comments to the Author” section, enter your conflict of interest statement in the “Confidential to Editor” section, and submit your "Accept" recommendation.

Reviewer #1: (No Response)

2. Is the manuscript technically sound, and do the data support the conclusions?

Reviewer #1: Yes

3. Has the statistical analysis been performed appropriately and rigorously? 

Reviewer #1: Yes

4. Have the authors made all data underlying the findings in their manuscript fully available?

Reviewer #1: Yes

5. Is the manuscript presented in an intelligible fashion and written in standard English?

Reviewer #1: No

6. Review Comments to the Author

Reviewer #1: In this revised version of the manuscript, the authors have addressed almost all of the points which I raised in my initial review. The link to their repository, which was not working in the original submission, has been replaced with a working link to a repository containing their code and data. The authors have also removed the references to policy which I identified as not being supported by the results of their study.

The only aspect of the paper which I feel is still insufficient Is the section on Pagerank index on pages 6 and 7. While this section is more detailed than it was previously, it still isn’t totally clear how Pagerank is calculated based on the details the authors provide. In particular, the authors do not define B_i, the set which they sum over in the formula at the top of page 7. However, given the authors have now provided a reference (Easley and Kleinberg 2010) which includes a detailed definition of Pagerank, I think it will be sufficient to give a qualitative description and point the reader to the reference for full detail. This description just needs to cover the basic idea that Pagerank measures how likely one is to arrive at a given node by moving randomly around a network, and that it is calculated iteratively.

While Figures 6 and 7 are useful for visualising the epidemic network, they are currently a bit small. Depending on the journal’s formatting requirements, it would be good to make these figures larger, possibly by rearranging each one into a 3x1 column of plots rather than the current 1x3 rows.

Minor grammar/typo notes:

• Abstract: “The spread of virus is highly related to the structure of human networks” should be “The spread of viruses is highly related to the structure of human networks”, or, since this holds for other types of pathogen, “The spread of infectious diseases is highly related to the structure of human networks”

• Abstract: “There are more male patients than the females in these two districts…” should be “There are more male patients than females in these two districts…”

• Top of page 5: “This study collects detailed information of patients diagnosed with COVID-19…” should be “This study collects detailed information about patients diagnosed with COVID-19…”

• Bottom of page 6: “By many times of iterations, the Pagerank value of each node in the network converge to a limiting value.”

• Top of page 10: “We also summarized the number of days that symptom had emerged”. I think the authors here mean “We also summarized the number of days between arrival in Hainan or Yunan and emergence of symptoms”, although the authors should confirm that this is indeed the intended meaning.

• Page 15: “However, these clusters did not form larger clusters in the late stage, indicating that most infected people were taken medical treatment or quarantined in time” should be “However, these clusters did not form larger clusters in the late stage, indicating that most infected people were given medical treatment or quarantined in time”

• Page 20: “When these information be quickly collected…” should be “When these information are quickly collected…”

• Page 20: “In general, most of the research on the COVID-19 epidemic focus on the epidemiological characteristics of confirmed patients” should be “In general, most of the research on the COVID-19 epidemic has focused on the epidemiological characteristics of confirmed patients”

• Page 21: “Because of limited date” should be “Because of limited data”.

7. PLOS authors have the option to publish the peer review history of their article (what does this mean?). If published, this will include your full peer review and any attached files.

Reviewer #1: **Yes: **Joe Hilton

---

## [Author Response · Author response to Decision Letter 1]

5 Oct 2021

Authors’ Response

Dear Editor and Reviewer,

Thank you very much for the targeted and constructive comments that you and the reviewer provided on our manuscript. We are very pleased with the detailed modification suggestions and grateful to have another opportunity to refine our paper. As you will see, we have addressed all the issues raised again, which are highlighted in yellow in the manuscript and detailed below point by point. We look forward to your rechecking the paper and making other valuable suggestions for us. 

Thank you once more for your constructive feedback and tireless patience. We hope that the refined manuscript meets your expectations.

Yours sincerely,

Authors of PONE-D-21-13537

Main comments: The only aspect of the paper which I feel is still insufficient Is the section on Pagerank index on pages 6 and 7. While this section is more detailed than it was previously, it still isn’t totally clear how Pagerank is calculated based on the details the authors provide. In particular, the authors do not define B_i, the set which they sum over in the formula at the top of page 7. However, given the authors have now provided a reference (Easley and Kleinberg 2010) which includes a detailed definition of Pagerank, I think it will be sufficient to give a qualitative description and point the reader to the reference for full detail. This description just needs to cover the basic idea that Pagerank measures how likely one is to arrive at a given node by moving randomly around a network, and that it is calculated iteratively.

Our response: Thank you very much for your attentiveness. We gave the define B_i in this section, where B_i is the set containing all nodes linking to the focal node j. We also have added a qualitative explanation of Pagerank that it measures how likely one is to arrive at a given node by moving randomly around a network, and it is calculated iteratively. We remind our readers that they can find the detailed calculation procedure in the book by Easley and Kleinberg.

Main comments: While Figures 6 and 7 are useful for visualising the epidemic network, they are currently a bit small. Depending on the journal’s formatting requirements, it would be good to make these figures larger, possibly by rearranging each one into a 3x1 column of plots rather than the current 1x3 rows.

Our response: Thanks for your suggestions. We have rearranged the plot of the pictures again to show more clear details of the contact network. Please see Figures 6 and 7.

Minor grammar/typo notes

Minor comments: Abstract: “The spread of virus is highly related to the structure of human networks” should be “The spread of viruses is highly related to the structure of human networks”, or, since this holds for other types of pathogen, “The spread of infectious diseases is highly related to the structure of human networks”.

Our response: Thank you very much for your suggestions for corrections. We have revised the sentence as “The spread of infectious diseases is highly related to the structure of human networks”.

Minor comments: Abstract: “There are more male patients than the females in these two districts…” should be “There are more male patients than females in these two districts…”.

Our response: Thank you very much for your suggestions for corrections. We have deleted the word “the” and revised the sentence as “There are more male patients than females in these two districts…”.

Minor comments: Top of page 5: “This study collects detailed information of patients diagnosed with COVID-19…” should be “This study collects detailed information about patients diagnosed with COVID-19…”.

Our response: Thank you very much for your corrections. We have adjusted the sentence as “This study collects detailed information about patients diagnosed with COVID-19…”.

Minor comments: Bottom of page 6: “By many times of iterations, the Pagerank value of each node in the network converge to a limiting value”.

Our response: Thank you. We have corrected the sentence as “The Pagerank value of all nodes in the network converge to limiting values as the number of iterations goes to infinity”.

Minor comments: Top of page 10: “We also summarized the number of days that symptom had emerged”. I think the authors here mean “We also summarized the number of days between arrival in Hainan or Yunan and emergence of symptoms”, although the authors should confirm that this is indeed the intended meaning.

Our response: Thank you very much for your correction. We have clarified the sentence as “We also summarized the number of days between arrival in Hainan or Yunan and emergence of symptoms”.

Minor comments: Page 15: “However, these clusters did not form larger clusters in the late stage, indicating that most infected people were taken medical treatment or quarantined in time” should be “However, these clusters did not form larger clusters in the late stage, indicating that most infected people were given medical treatment or quarantined in time”

Our response: Thank you very much. We have changed this sentence as: “However, these clusters did not form larger clusters in the late stage, indicating that most infected people were given medical treatment or quarantined in time”.

Minor comments: Page 20: “When these information be quickly collected…” should be “When these information are quickly collected…”

Our response: Thank you very much for your corrections. We have modified the sentence into “When these information are quickly collected…”.

Minor comments: Page 20: “In general, most of the research on the COVID-19 epidemic focus on the epidemiological characteristics of confirmed patients” should be “In general, most of the research on the COVID-19 epidemic has focused on the epidemiological characteristics of confirmed patients”

Our response: Thank you. We have revised the sentence into “In general, most of the research on the COVID-19 epidemic has focused on the epidemiological characteristics of confirmed patients”.

Minor comments: Page 21: “Because of limited date” should be “Because of limited data”.

Our response: Thank you. We corrected the mistakes in the sentence.

Our response: Moreover, we checked the full manuscript again and corrected some other grammatical errors. Thank you again for your very detailed and extremely constructive comments, they help improve this paper a lot. We would not have got a high-quality paper without these sincere suggestions. We look forward to hear more about your feedback.

---

## [Decision Letter · Decision Letter 2]

25 Oct 2021

PONE-D-21-13537R2Contact Network Analysis of Covid-19 in Tourist Areas——Based on 333 Confirmed Cases in ChinaPLOS ONE

Dear Dr. Wang,

Thank you for submitting your manuscript to PLOS ONE. After careful consideration, we feel that it has merit but does not fully meet PLOS ONE’s publication criteria as it currently stands. Therefore, we invite you to submit a revised version of the manuscript that addresses the points raised during the review process. Overall, you have addressed most of the reviewer's comments. In the revised version, please ensure you address all the remaining issues.

We look forward to receiving your revised manuscript.

Kind regards,

Federico Botta

Academic Editor

PLOS ONE

Journal Requirements:

Reviewers' comments:

Reviewer's Responses to Questions

**Comments to the Author**

1. If the authors have adequately addressed your comments raised in a previous round of review and you feel that this manuscript is now acceptable for publication, you may indicate that here to bypass the “Comments to the Author” section, enter your conflict of interest statement in the “Confidential to Editor” section, and submit your "Accept" recommendation.

Reviewer #1: All comments have been addressed

2. Is the manuscript technically sound, and do the data support the conclusions?

Reviewer #1: Yes

3. Has the statistical analysis been performed appropriately and rigorously? 

Reviewer #1: Yes

4. Have the authors made all data underlying the findings in their manuscript fully available?

Reviewer #1: Yes

5. Is the manuscript presented in an intelligible fashion and written in standard English?

Reviewer #1: No

6. Review Comments to the Author

Reviewer #1: Main comments

The paper has overall been corrected to a good standard. In particular the section on PageRank, which I identified as lacking in my previous round of comments, has been amended and is now sufficient pending a few comments on grammar which I have noted below.

I have two further notes, which I did not catch on my previous rounds of comments. In the abstract the authors state that there are more male than female patients, whereas Table 1 (pages 7 to 8) and the accompanying text on page 7 show that there are 142 female patients to 140 male patients. The abstract text should be corrected accordingly.

Further, on page 9 the authors say “In the data of Hainan Province, except for January 27, January 28, and February

9, the average age is stable at 40-60 years old, that is, these cases are older people

with a weakened immune system”. This is a fairly sweeping statement. First of all, the authors do not indicate what the mean age of the underlying population is – if the mean population age is in the 40-60 years old range, then this average case age is what one would expect with no underlying age structure to transmission or symptom intensity. If 40-60 is actually older than the baseline population average, then there is still no direct evidence that the high number of cases in older individuals is caused by older individuals having weakened immune systems. This is simply one hypothetical explanation, and the authors should present it as a possibility rather than a certainty. On page 22 the authors then say “At the late stage, cases

are mostly elderly people who have a weak immune system, poor mobility ability, and weak transmission ability”. This should be modified to clarify that these are all risk factors associated with old age, but that not every case will exhibit all of these risk factors (and some may exhibit none of them).

Minor comments

Page 1: “On January 30, 2020, The World Health Organization declared the

epidemic as a Public Health Emergency of International Concern.” – the (newly added) word “as” is not necessary here.

Page 4: “Human social network is heterogeneous” should be “Human social networks are heterogeneous”.

Page 5: “The specific steps of text mining and coding are as following:” should be “The specific steps of text mining and coding are as follows.” Note that since that the sentence is followed by a paragraph break rather than a bulleted list, the authors should replace the semi colon with a full stop.

Page 6: “PageRank measures how likely one is to arrive at a given node by moving randomly around a network, and that it is calculated iteratively” should be “PageRank

measures how likely one is to arrive at a given node by moving randomly around a

network, and is calculated iteratively”.

Page 7: “The specific formula is as following” should be “The specific formula is as follows”.

Page 7: “A component means that, within in it, all

members can be conneted by a well-defined path.” – “connected” is misspelt.

Page 9: “One possible reason for the age gap might be that tourist industry is the pillar industry in Hainan Province” should be “One possible reason for the age gap might be that tourism is the pillar industry in Hainan Province”

Page 9: “As for Yunnan Province, though it is also a tourist attraction, it attracts people at all ages. Besides, Yunnan is one of the biggest labor exporters in China, these returning migrant workers, who are mostly are young, also contributes to the lower the average age of Yunnan.” The phrasing of the first sentence feels slightly informal for a scientific paper. Something like “Tourism in Yunan Province is typically less concentrated towards older age groups” would be a bit more fitting. I do not find the second sentence at all convincing – being a net exporter of migrant labour will make the average age higher, not lower. Returning workers will bring the age profile back up to the pre-migration average but will not increase it past that level. Unless I have seriously misunderstood the reasoning here, this statement should be removed.

Pages 11-12: “For example, cases No. 73 and No. 87 in Hainan Province crossed districts and counties for about 8 times” should just be “For example, cases No. 73 and No. 87 in Hainan Province crossed districts and counties about 8 times”.

Page 19: “maximum” is misspelt as “maximun”.

Page 24: “In such a condition” should be “Under such conditions”.

7. PLOS authors have the option to publish the peer review history of their article (what does this mean?). If published, this will include your full peer review and any attached files.

Reviewer #1: **Yes: **Joe Hilton

---

## [Author Response · Author response to Decision Letter 2]

31 Oct 2021

Authors’ Response

Dear Reviewer,

Thank you very much for the targeted and constructive comments that you and the reviewer provided on our manuscript. We are very pleased with the detailed modification suggestions and grateful to have another opportunity to correct errors and refine our work. As you will see, we have addressed all the issues raised again, which are highlighted in yellow in the manuscript and detailed below point by point. We look forward to your rechecking the paper and making other valuable suggestions for us. 

Thank you once more for your constructive feedback and tireless patience. We hope that the refined manuscript meets your expectations.

Yours sincerely,

Authors of PONE-D-21-13537

Main comments: In the abstract the authors state that there are more male than female patients, whereas Table 1 (pages 7 to 8) and the accompanying text on page 7 show that there are 142 female patients to 140 male patients. The abstract text should be corrected accordingly.

Our response: Thank you very much for your attentiveness. We have compared the data in Table 1, that is, the number of female patients is more than that of male patients, and revised the expression in the abstract.

Main comments: Further, on page 9 the authors say “In the data of Hainan Province, except for January 27, January 28, and February 9, the average age is stable at 40-60 years old, that is, these cases are older people with a weakened immune system”. This is a fairly sweeping statement. First of all, the authors do not indicate what the mean age of the underlying population is – if the mean population age is in the 40-60 years old range, then this average case age is what one would expect with no underlying age structure to transmission or symptom intensity. If 40-60 is actually older than the baseline population average, then there is still no direct evidence that the high number of cases in older individuals is caused by older individuals having weakened immune systems. This is simply one hypothetical explanation, and the authors should present it as a possibility rather than a certainty.

Our response: Thanks for your suggestions. In this section, we increased the proportions of people aged 60 and above among the cases in Yunnan and Hainan Province, and compared them with the proportions of people of this age in these two provinces shown in the seventh National Census of China. By comparing the difference of the proportion of the elderly in the patients and the census data of the whole province, we find that the proportion of the elderly in the patients is higher, which means that the elderly may have a larger proportion in the confirmed patients due to weak immunity and other reasons. Therefore, we revised the statement in the paper to “Tourism in Yunnan Province is typically less concentrated towards older age groups. In Yunnan Province, 21.01% of the cases are aged 60 years and above, which exceeds the proportion of the local population aged 60 years and above in the total population (14.91%, China the seventh National Census). In Hainan Province, 32.72% of the cases are aged 60 years and above, which is greater than the proportion of this age group in the total population (14.65%, China the seventh National Census).”.

Main comments: On page 22 the authors then say “At the late stage, cases

are mostly elderly people who have a weak immune system, poor mobility ability, and weak transmission ability”. This should be modified to clarify that these are all risk factors associated with old age, but that not every case will exhibit all of these risk factors (and some may exhibit none of them).

Our response: Thanks for your suggestions. We have modified to clarify that higher age is likely to cause patients to who have a weak immune system, poor mobility ability, and weak transmission ability.

Minor grammar/typo notes

Minor comments: Page 1: “On January 30, 2020, The World Health Organization declared the epidemic as a Public Health Emergency of International Concern.” – the (newly added) word “as” is not necessary here.

Our response: Thank you very much for your suggestions for corrections. We have deleted the word “as” in the sentence and revised the sentence as “On January 30, 2020, The World Health Organization declared the epidemic a Public Health Emergency of International Concern”.

Minor comments: Page 4: “Human social network is heterogeneous” should be “Human social networks are heterogeneous”.

Our response: Thank you very much for your suggestions for corrections. We have revised the sentence as “Human social networks are heterogeneous”.

Minor comments: Page 5: “The specific steps of text mining and coding are as following:” should be “The specific steps of text mining and coding are as follows.” Note that since that the sentence is followed by a paragraph break rather than a bulleted list, the authors should replace the semi colon with a full stop.

Our response: Thank you very much for your corrections. We have adjusted the sentence as “The specific steps of text mining and coding are as follows”.

Minor comments: Page 6: “PageRank measures how likely one is to arrive at a given node by moving randomly around a network, and that it is calculated iteratively” should be “PageRank measures how likely one is to arrive at a given node by moving randomly around a network, and is calculated iteratively”.

Our response: Thank you. We have corrected the sentence as “PageRank measures how likely one is to arrive at a given node by moving randomly around a network, and is calculated iteratively”.

Minor comments: Page 7: “The specific formula is as following” should be “The specific formula is as follows”.

Our response: Thank you very much for your correction. We have clarified the sentence as “The specific formula is as follows”.

Minor comments: Page 7: “A component means that, within in it, all members can be conneted by a well-defined path.” – “connected” is misspelt.

Our response: Thank you very much. We corrected the misspelt word in the sentence.

Minor comments: Page 9: “One possible reason for the age gap might be that tourist industry is the pillar industry in Hainan Province” should be “One possible reason for the age gap might be that tourism is the pillar industry in Hainan Province”

Our response: Thank you very much for your corrections. We have modified the sentence into “One possible reason for the age gap might be that tourism is the pillar industry in Hainan Province”.

Minor comments: Page 9: “As for Yunnan Province, though it is also a tourist attraction, it attracts people at all ages. Besides, Yunnan is one of the biggest labor exporters in China, these returning migrant workers, who are mostly are young, also contributes to the lower the average age of Yunnan.” The phrasing of the first sentence feels slightly informal for a scientific paper. Something like “Tourism in Yunan Province is typically less concentrated towards older age groups” would be a bit more fitting. I do not find the second sentence at all convincing – being a net exporter of migrant labour will make the average age higher, not lower. Returning workers will bring the age profile back up to the pre-migration average but will not increase it past that level. Unless I have seriously misunderstood the reasoning here, this statement should be removed.

Our response: Thank you. We have revised the sentence into “Tourism in Yunan Province is typically less concentrated towards older age groups”. We also removed the impertinent statement in the paragraph

Minor comments: Pages 11-12: “For example, cases No. 73 and No. 87 in Hainan Province crossed districts and counties for about 8 times” should just be “For example, cases No. 73 and No. 87 in Hainan Province crossed districts and counties about 8 times”.

Our response: Thank you. We corrected the mistakes and revised the sentence into “For example, cases No. 73 and No. 87 in Hainan Province crossed districts and counties about 8 times”.

Minor comments: Page 19: “maximum” is misspelt as “maximun”.

Our response: Thank you. We corrected the mistake in the sentence and spelled the word “maximun” again.

Minor comments: Page 24: “In such a condition” should be “Under such conditions”.

Our response: Thank you. We corrected the mistake and revised the sentence into “Under such conditions”.

Our response: Moreover, we checked the full manuscript again and corrected some other grammatical errors. Thank you again for your very detailed and extremely constructive comments. They help us to improve this paper a lot. We would not have got a high-quality paper without these sincere suggestions. We look forward to your feedback.

---

## [Decision Letter · Decision Letter 3]

1 Dec 2021

Contact Network Analysis of Covid-19 in Tourist Areas——Based on 333 Confirmed Cases in China

PONE-D-21-13537R3

Dear Dr. Wang,

We’re pleased to inform you that your manuscript has been judged scientifically suitable for publication and will be formally accepted for publication once it meets all outstanding technical requirements.

Kind regards,

Federico Botta

Academic Editor

PLOS ONE

Additional Editor Comments (optional):

Reviewers' comments:

Reviewer's Responses to Questions

**Comments to the Author**

1. If the authors have adequately addressed your comments raised in a previous round of review and you feel that this manuscript is now acceptable for publication, you may indicate that here to bypass the “Comments to the Author” section, enter your conflict of interest statement in the “Confidential to Editor” section, and submit your "Accept" recommendation.

Reviewer #1: All comments have been addressed

2. Is the manuscript technically sound, and do the data support the conclusions?

Reviewer #1: Yes

3. Has the statistical analysis been performed appropriately and rigorously? 

Reviewer #1: Yes

4. Have the authors made all data underlying the findings in their manuscript fully available?

Reviewer #1: Yes

5. Is the manuscript presented in an intelligible fashion and written in standard English?

Reviewer #1: Yes

6. Review Comments to the Author

Reviewer #1: The authors have addressed all of my comments to a good standard, and I feel that the manuscript is now ready for publication.

7. PLOS authors have the option to publish the peer review history of their article (what does this mean?). If published, this will include your full peer review and any attached files.

Reviewer #1: **Yes: **Joe Hilton

---

## [Editor Report · Acceptance letter]

3 Dec 2021

PONE-D-21-13537R3 

Contact Network Analysis of Covid-19 in Tourist Areas——Based on 333 Confirmed Cases in China 

Dear Dr. Wang:

I'm pleased to inform you that your manuscript has been deemed suitable for publication in PLOS ONE. Congratulations! Your manuscript is now with our production department. 

Kind regards, 

on behalf of

Dr. Federico Botta 

Academic Editor

PLOS ONE